# Resonant Coupling of Hermite-Gaussian Transverse Modes in the Triangular Cavity of a Cavity Ring-down Spectroscope

Yijie Ren [1,2], Changxiang Yan [1,3], Xiamiao Zhang [4] and Chunhui Hu [1,*]

1 Changchun Institute of Optics, Fine Mechanics and Physics, Chinese Academy of Sciences, Changchun 130033, China
2 University of Chinese Academy of Sciences, Beijing 100049, China
3 Center of Materials Science and Optoelectrics Engineering, University of Chinese Academy of Sciences, Beijing 100049, China
4 Changchun JiaCheng Information Technology, Changchun 130103, China
* Correspondence: famous226@163.com

**Abstract:** During resonance in resonant cavities, such as those used in laser or cavity ring-down spectroscopes (CRDS), resonant coupling between higher-order transverse modes and fundamental modes can seriously affect the quality of the beam and introduce measurement errors. Several coupling models, such as thermal deformation coupling and scattering coupling, have been established according to existing coupling theory and specific application scenarios; however, these coupling models have not been attributed to a unified theory. In this paper, we reveal that the same resonant coupling excitation factors exist under different types of environmental perturbation. The conditions and range of resonant coupling in a CRDS ring-down cavity are systematically analyzed, and a preferential coupling model of the middle-order modes is proposed. The time-domain characteristics of the CRDS are used in experiments to analyze the resonant coupling between the modes in a weak energy system. The order and coupling range of the middle-order modes involved in resonant coupling are verified using the modal filtering characteristics of the triangular cavity; this paper presents a unified explanation for various types of resonant coupling and also provides a new approach to resonant coupling experiments performed in high-finesse resonant cavities.

**Keywords:** CRDS; absorption spectroscopy; laser mode; resonant coupling; triangular cavity

## 1. Introduction

Since the invention of the cavity ring-down spectrometer (CRDS), it has been used in terrestrial environments and space remote to measure trace gases quickly and accurately in complex environments [1–3]. Absorption Spectroscopy Instruments has undergone a series of technical innovations, such as improvements to its laser light source and detector level, allowing it to be more widely applied [4,5]. The core component is the ring-down cavity, for which a triangular cavity is more often used than a conventional straight cavity as it better suppresses optical feedback [6]. Therefore, understanding the characteristics of triangular cavities is of great importance for the improvement of CRDS instruments.

Triangular resonant cavities are heavily studied and used as a mode cleaner in gravitational wave detection. Due to their ability to filter three major properties simultaneously—spectral, polarization, and spatial—they are used for outputting narrow-linewidth, single-polarization, and single-mode laser beams. Kane studied the spectral filtering characteristics of triangular resonant cavities [7], while Saraf analyzed their polarization filtering characteristics and verified them experimentally [8]; however, research on the mode cleaning characteristics of triangular cavities remains at the experimental level [9,10]. The filtering principle of single-space modes and the different coupling mechanisms of low- and high-order modes have not been studied in depth. Here, the low- and high-order modes are commonly referred

to as the Hermite–Gaussian (HGS) mode, which is a generic term for non-fundamental modes.

In the field of CRDS, coupling of a fundamental mode Gaussian beam with an HGS mode leads to cavity lifetime anomalies and introduces measurement errors. In the study of spatial modes in resonant cavities by HGS function approximation or Laguerre-Gaussian approximation, the spatial modes can be divided into fundamental, low-order, and high-order modes. There are two explanations for the coupling of HGS modes: coherent coupling [11] and incoherent coupling [12–14]. In Klaassen's analysis, the coherent cause can better explain the HGS mode coupling phenomenon in the cavity, while the coupling cause is attributed to scattering coupling [11]. Amber attributed multimode coupling to resonant coupling due to thermal deformation [15]. Courtois suggested that resonant coupling is caused by astigmatism due to curvature error of the cavity reflector [16].

The main cause of mode coupling is usually explained as being due to resonant coupling between the fundamental mode and HGS mode based on the coherence principle; however, there is no uniform explanation for the cause of resonant coupling; it is not known whether the scattering coupling model, thermal coupling model, and astigmatism model are related. We believe that resonant coupling in different scenarios must contain the same causal factors; however, such a universally applicable theoretical analysis has not been reported. The difference in the order of HGS modes when resonant coupling occurs in different application environments, such as laser resonant cavities, gravitational wave mode cleaners, and CRDS ring-down cavities, is unknown. Nor is it known whether the order of HGS modes will be preferentially resonantly coupled in a CRDS. The factors affecting the frequency shift range of HGS modes when coupling occurs have not been systematically investigated.

In this paper, based on the independence of the dynamic response operators of the frequency, mode, and polarization of a triangular resonant cavity, the modal filtering characteristics and the polarization filtering characteristics of a triangular cavity are analyzed. In resonant coupling analysis, the various types of environmental perturbations can be attributed to the variables $\Delta R$ and $\Delta L$. The low-order and high-order modes near the resonant frequency of the fundamental mode are not preferentially coupled due to various complex factors, such as the drift range of the $g$-factor and diffraction limitations. Preferential coupling of the middle-order modes in a CRDS system is proposed for the first time. An experiment is used to verify the resonant coupling of the middle-order modes by the time-domain method based on the unique filtering characteristics of the triangular cavity. The order and frequency drift range of the middle-order modes where resonant coupling occurs are also determined.

Based on the filtering characteristics of the CRDS triangular cavity, this paper analyzes and verifies the influence of resonant coupling in different application environments. The modal and polarization filtering characteristics of the triangular cavity are focused on in Section 2. In Section 3, the key variables of resonant coupling, $\Delta R$ and $\Delta L$, are analyzed based on a coupling matrix, and the frequency offset range of the HGS mode at the time of coupling is explored. The resonant coupling of the middle-order modes was verified via the experiments described in Section 4, and the order and coupling range of the middle-order modes are discussed in Section 5. The final conclusions are given in Section 6.

## 2. Triangular Cavity Dynamic Response Operator

The modal and polarization properties of the filtering characteristics are the keys to distinguishing a triangular cavity from other types of cavities. Research on the dynamic response characteristics of the electric field in the cavity was conducted by Anderson et al. for HGS modes [17,18] and by Sigg for a more comprehensive dynamic response propagation operator [19]. The study of triangular cavity mode and polarization filtering characteristics according to the approximate theory for mode coupling is the key to experimentally verifying the HGS mode coupling mechanism; this section explains the independence of the three main characteristics, describes the principles of mode separation

and polarization phase shift in the triangular cavity in detail, and gives an expression of the propagation factor acting on the total electric field. A basis is provided for the verification of an amplitude coupling model for resonant coupling theory.

### 2.1. Independence between the Filter Characteristics of Triangular Resonant Cavities

In analyzing a triangular cavity based on HGS theory, due to the modal resonant frequency separation property of the triangular cavity, the electric field within it can be expressed as [20]:

$$
\begin{aligned}
E\left(x, y, q\left(z_{(j)}\right)\right) = & \sum_{m,n} a_{mn} H_m\left(\frac{\sqrt{2}}{\omega_{(q(z(j)))}}x\right) H_n\left(\frac{\sqrt{2}}{\omega_{(q(z(j)))}}y\right) \times \left(\frac{\omega_0}{\omega_{(q(z(j)))}}\right) \\
& \times \exp\left(-i\left(m+\tfrac{1}{2}\right)\phi_m\left(q\left(z_{(j)}\right)\right) - \frac{x^2}{\omega^2_{(q(z(j)))}}\right) \\
& \times \exp\left(-i\left(n+\tfrac{1}{2}\right)\phi_n\left(q\left(z_{(j)}\right)\right) - \frac{y^2}{\omega^2_{(q(z(j)))}}\right)
\end{aligned}
\tag{1}
$$

where $a_{mn}$ represents the complex amplitude, $\omega_0$ is the beam radius of the fundamental mode, $q(j)$ is the complex beam parameter in the $j$th segment, $m$ and $n$ are the orders of the HGS modes, $H_m$ is a Hermite polynomial, $\omega_{(q(z(j)))}$ is the beam radius of the fundamental mode at position $z_{(j)}$, and $\phi_m$ and $\phi_n$ represent the Gouy phases in the meridional and sagittal planes, respectively.

A complete electric field vector expression that includes frequency and polarization information can provide generality in analyzing the detector response and noise, even in more complex systems. The periodic frequency component in Equation (1) and the polarization component containing the vibration information are included in Equation (2); both equations are under paraxial approximation [17].

$$
E(x,y,z,t) = \sum_{mn}\sum_{r}\sum_{p} a_{mnrp}\exp[i(\omega_o + r\omega)t]U_{mn}(x,y,z)\varepsilon_{polariza}
\tag{2}
$$

where $U_{mn}(x,y,z)$ is the Hermite orthogonal expression in Equation (1), $a_{mnrp}$ represents the amplitude containing the mode, frequency and polarization information, $\omega_0$ is the angular frequency of the laser light, $\omega$ is the angular frequency of the variation, and $\varepsilon$ contains s- and p-polarization information.

In a passive cavity system such as a CRDS, the dynamic propagation expression for the resonance established by the injected field $E_{in}$ can be expressed as:

$$
E_{cav} = G_{roundtrip}E'_{cav} + T_1 E_{in}
\tag{3}
$$

where $E_{cav} = E'_{cav}$ denotes the static field conditions after stability is reached, and $G_{roundtrip}$ is similar in meaning to the conventional *ABCD* propagation matrix, which contains the reflection and propagation operators of the circulating field in the triangular cavity $G_{roundtrip} = P_1 M_2 P_2 M_3 P_3 M_1$. The effects of the modal and polarization properties of the triangular cavity on these operators are described in detail in the next two sections.

### 2.2. Resonant Frequency Separation of Same-Order HGS Modes

The modal separation property of the triangular cavity can be expressed as an additional rotation factor of the reflection operator of Equation (4). Figure 1 shows that the modal pattern is constant in the $y$-direction and changes in the $x$-direction after a single reflection; $x$, $m$ represent the direction parallel to the plane of the triangular cavity, while $y$, $n$ represent the direction perpendicular to the plane of the triangular cavity. The straight cavity has its modal pattern reunited after two reflections. The triangular cavity, on the other hand, cannot achieve reunion after three reflections, so the modal pattern in the $x$-$z$ plane needs to be added with a rotation factor $b$ related to the number of reflections. When $m$ is an even-order mode, the symmetry is not affected by the modal non-reconvergence.

$$M_1' = (-1)^{mb}M_1 \tag{4}$$

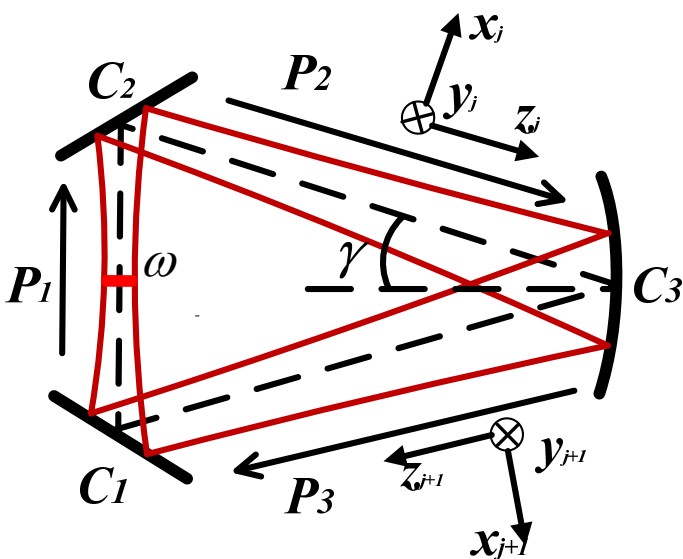

**Figure 1.** Pattern matching of the triangular ring cavity. $C_1$ and $C_2$ are plane mirrors, $C_3$ is concave mirror. The radius of curvature of the concave mirror is R. Let the length of the closed optical path in the resonant cavity be $P = P_1 + P_2 + P_3$, and $\gamma$ represents the incidence angle of $C_3$, and $\omega$ represents the laser beam waist. $x_{(j+1)}, y_{(j+1)}, z_{(j+1)}$ means $x_{(j)}, y_{(j)}, z_{(j)}$ has undergone a reflection.

The degenerate condition [21] for the HGS mode of the triangular cavity is:

$$kL - (m + n + 1)\arccos(\sqrt{g}) - \frac{\pi(1 - (-1)^m)}{2} = 2\pi q. \tag{5}$$

where $k = \omega/c$ corresponds to the wavenumber, $L$ represents the cavity length of the triangular cavity, $g$ is the structural factor of the triangular cavity $g = 1 - L/2R, f = R/2, R$ is the curvature of $C_3$, and $q$ is the longitudinal mode number. Thus, the resonant frequency $\nu_{mn}$ of each order of mode can be expressed as:

$$\nu_{mnq} = \begin{cases} FSR \times q + \frac{FSR}{\pi}(m + n + 1)\arctan(\sqrt{1 - \frac{L}{R}}) \text{even cavity} \\ FSR \times q + \frac{FSR}{\pi}(m + n + 1)\arctan(\sqrt{1 - \frac{L}{R}}) \text{odd cavity} \\ \quad + \frac{FSR}{2}\frac{(1 - (-1)^m)}{2} \end{cases} \tag{6}$$

where $FSR$ is the free spectral range. The transmission factor of the triangular cavity can be expressed as Equation (7) [22]. The resonant frequency separation of the odd-order modes of the triangular cavity is shown in Figure 2 as an example for the first-order and second-order modes. $T_{00}$ is the transmittance of the fundamental mode, and $F$ is the finesse.

$$T_{mn} = T_{00} \frac{1}{\left[1 + \frac{4F^2}{\pi^2}\sin^2((m + n)\arccos(\sqrt{1 - \frac{L}{R}}) + \frac{\pi(1 - (-1)^m)}{4})\right]^{1/2}}. \tag{7}$$

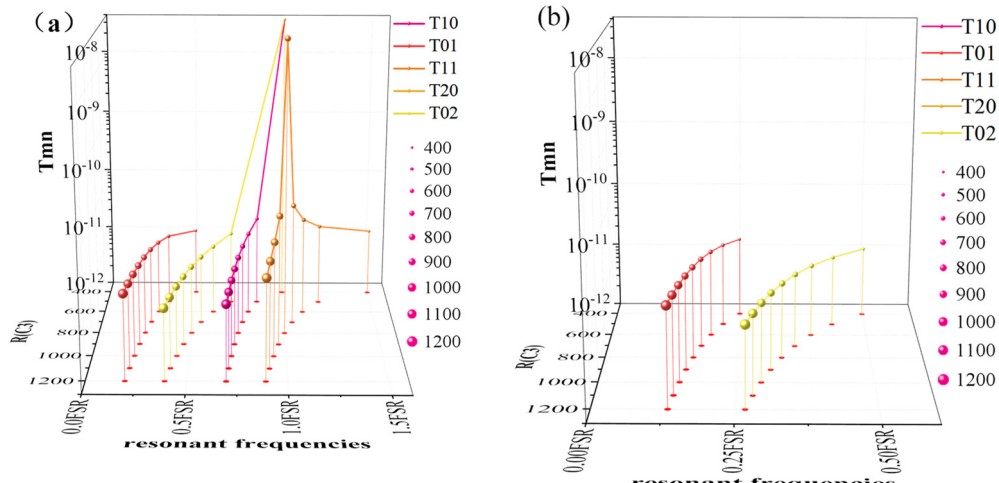

**Figure 2.** First-order and second-order mode transmittance of a Gaussian laser beam for different end-mirror radii in a (**a**) triangular ring-shaped cavity and (**b**) straight cavity.

Astigmatism is considered as a possible factor for modal coupling. Astigmatism in a straight cavity leads to the coupling of $\text{TEM}_{01}$ and $\text{TEM}_{10}$ [16]. The role of astigmatism in a triangular cavity is given by Equation (8). In a triangular cavity, the astigmatism is determined by angle $\gamma$ in Figure 1, which is attached to the $m$ and $n$ modes as the astigmatism factor. The frequency interval of the triangular cavity's astigmatic factor is generated on the $m,n$ modes, respectively, and the magnitude is much smaller than the frequency interval generated by the modes. For example, the difference between $\text{TEM}_{01}$ and $\text{TEM}_{10}$ is $FSR/2$, and the effect of astigmatism can be excluded in the triangular cavity experiment.

$$v_{mnq}^{\text{ast}} = FSR \times \left( q + \frac{1}{2\pi}\left(m + \frac{1}{2}\right) \times \arccos\left(\sqrt{1 - \frac{L}{R\cos\gamma}}\right) + \frac{1}{2\pi}\left(n + \frac{1}{2}\right) \right.$$
$$\left. \times \arccos\left(\sqrt{1 - \frac{L\cos\gamma}{R}}\right) + \frac{\pi(1-(-1)^m)}{4} \right) \tag{8}$$

### 2.3. Polarization Phase Shift of the Triangular Cavity

The phase shift difference generated by the reflection between two polarizations is another property of triangular cavities. The Fresnel coefficient describes the reflection coefficient between the two polarization states. When the reflection angle is less than the Brewster angle, the *s*- and *p*-light will produce a phase difference of $\pi$.

$$r_s = -\frac{\sin(\theta_i - \theta_t)}{\sin(\theta_i + \theta_t)}$$
$$r_p = \frac{\tan(\theta_i - \theta_t)}{\tan(\theta_i + \theta_t)} \tag{9}$$

Therefore, the polarization phase difference can be understood as the difference in resonance length [8]:

$$L_p = 2\pi c/\omega$$
$$L_s = 2\pi c/\omega_s + \pi c/\omega \tag{10}$$

where $\omega$ is the incident light frequency, $c$ is the speed of light, and $L_s$ and $L_p$ are the resonant round-trip lengths of the cavity for *s*-polarized and *p*-polarized light, respectively. Thus, the additional operator for the polarization characteristics of the triangular cavity can be expressed as:

$$\varepsilon' = \begin{bmatrix} r_p & 0 \\ 0 & r_s \end{bmatrix} \begin{bmatrix} -1 & 0 \\ 0 & 1 \end{bmatrix}^b \varepsilon \tag{11}$$

where $\varepsilon$ and $\varepsilon'$ represent the polarization information of *s*- and *p*-polarized light before and after reflection, respectively, and *b* represents the number of cavity mirrors.

After the analysis in this section, we can conclude that in the triangular cavity (*b* = 3), both the mode spacing and polarization characteristics will produce a resonant frequency difference between the meridional and sagittal planes, which will lead to a triangular cavity having a larger number of degenerate states in a free spectral range than a straight cavity. Different polarization states of the same order mode will have different resonant frequencies. When *m* is an odd number, a frequency spacing of *FSR*/2 will be generated between modes of the same order; this will greatly facilitate our experimental determination of the order and frequency shift of the HGS modes when modal coupling occurs.

## 3. HGS Transverse Mode Coupling Theory

The *q*-parameters of the Gaussian beam inside each resonant cavity are described by a deterministic *ABCD* matrix. The *q*-parameters contain information such as wavefront curvature $\rho$ and beam waist size $\omega_b$. The field transport matrix, refined in Section 2, contains the complete modal and polarization information of the beam inside the cavity and can be used to analyze the coupling of the modes. We refer to the Gaussian beam mass damage induced by external environmental changes, such as thermal lensing, thermal deformation, mirror damage, and external vibrations, as environmental perturbations. The purpose of this section is to study the same causal factors under the different environmental disturbances. Coupling models adapted to various resonant cavity application scenarios are derived and predictions of HGS modal information are made where coupling occurs.

### 3.1. Establishment and Approximation of Mode Coupling Theory

Initial research on the coupling of each higher-order mode to the fundamental mode focused on the deterioration in laser beam quality. The thermal lensing effect in the laser resonant cavity leads to interactions between the modes of each order, which eventually degrade the beam quality. The initial studies attributed this to incoherent interactions [13,14]. When an environmental perturbation occurs, it first causes coupling between the complex amplitude coefficients of different cavity modes [12]. The most dominant coupling process is that the distortion caused by environmental perturbation couples a certain degree of the amplitude of the fundamental mode to the higher order modes; however, it can be seen from Equation (7) that the resonant frequencies of the fundamental mode and higher-order mode are not the same and, even though the external perturbation causes modal coupling, this non-coherent coupling does not accumulate continuously.

Resonant coupling theory can more rationally explain the high-intensity coupling of higher-order modes found in lasers and CRDS. The resonant excitation of such higher-order modes not only deteriorates the laser beam quality [10] but also seriously affects the fitting accuracy of the ring-down curves [23]. Klaassen established a resonant coupling model between multiple modes [11], where *C* represents the coupling factor of each order of HGS mode:

$$M = \begin{bmatrix} C_{00} & C_{01} & \dots \\ C_{10} & \varsigma + C_{11} & \dots \\ \dots & \dots & \dots \end{bmatrix} \tag{12}$$

where $\varsigma$ is the frequency detuning away from degeneracy. Different systems have different *C* values. In the CRDS application scenario, the ring-down cavity finesse is up to $10^4$ orders of magnitude, and the linewidth of the laser in the cavity is 10 kHz. The fundamental mode linewidth in CRDS is much lower than the resonant linewidth of most lasers. The ring-down curves from CRDS experiments are fitted to a double-e exponential model [23], with a simplified model of coupling a single HGS mode to the fundamental mode being more suitable for the analysis of narrow linewidth ring-down cavities [24].

$$\begin{pmatrix} a'_0 \\ a'_1 \end{pmatrix} = \begin{pmatrix} 1 & 0 \\ 0 & Ae^{i\Delta} \end{pmatrix} \begin{pmatrix} \sqrt{1-|C|^2} & -C^* \\ C & \sqrt{1-|C|^2} \end{pmatrix} \begin{pmatrix} a_0 \\ a_1 \end{pmatrix}$$
$$= \begin{pmatrix} \sqrt{1-|C|^2} & -C^* \\ Ae^{i\Delta}C & Ae^{i\Delta}\sqrt{1-|C|^2} \end{pmatrix} \begin{pmatrix} a_0 \\ a_1 \end{pmatrix} \tag{13}$$

where $a_0$ and $a_1$ are the corresponding complex amplitudes of the fundamental and higher-order modes, respectively, $C$ is the coupling strength, and $A$ is the loss factor. $\Delta$ is the phase shift of the higher-order mode relative to the fundamental mode for each round trip. The first matrix represents the loss and coupling conditions of the higher-order modes where the coupling occurs. The second matrix represents the coupling strength of the higher-order modes.

The coupling condition derived from the total matrix is the $\Delta$ phase shift and will be the focus of study in the second part of this section. The frequency offset range of the HGS mode when coupling occurs depends on the loss factor of the total matrix and the coupling strength, which will be discussed in Section 3.3.

### 3.2. Resonance Coupling Causes

Among the wide range of externally induced coupling models that have been widely analyzed, such as scattering, thermal deformation, and pressure deformation models, some are applicable to strong energy systems such as lasers and some are applicable to weak energy systems such as ring-down cavities. Different coupling models are applicable to different cavity structures. Therefore, it is important to obtain a unified coupling mechanism to improve the coupling theory.

The triangular cavity we use is shown in Figure 1 and can be equated to a double-mirror cavity consisting of plane and curved mirrors. The beam waist in the cavity is located at the plane mirror. When the cavity is subjected to environmental perturbation, the $g$-factor causes a change in the Gouy phase, which can be expressed as $\phi = \cos^{-1}\left(\sqrt{g_{1,\text{dis}} \times g_{2,\text{dis}}}\right)$. In an ideal triangular cavity CRDS system that is not subjected to environmental perturbation, the cavity length is assumed to be 420 mm, the $C_3$ radius of curvature is 1 m, $L_1 = L_3 = 200$ mm, and $L_2 = 20$ mm. The frequency spacing between each order mode and the fundamental mode in the triangular resonant cavity is obtained from Equation (6), as shown in Figure 3b. The blue dots in the figure are the even-order modes within orders 1–180, and the red dots are the frequency spacing between the odd modes and fundamental mode. Two frequency spacings exist for odd modes of the same order within a triangular cavity [22]. Figure 3a shows the frequency spacing of the HGS mode in a straight cavity.

The HGS mode undergoes frequency drift when the environment is disturbed, and the offset rates of different order modes are different. When a higher order mode has the same resonant frequency as the fundamental mode, the condition $\Delta\nu = 0$ is satisfied, where $\Delta\nu_T$ is the transverse mode spacing, and $m$ and $n$ have different transverse mode spacing in the triangular cavity.

$$\Delta\nu = \Delta(m+n)\Delta\nu_T - q \times FSR \tag{14}$$

When the concave mirror $C_3$ introduces much larger ambient vibrations than the plane mirror, $g_1 = 1$ and the perturbation of the $g$-factor of the triangular cavity can be expressed as Equation (15).

$$g_1 g_{2,\text{dis}} = 1 - \frac{\frac{L_{\text{tri,dis}}}{2}}{R_{M3,\text{dis}}} \tag{15}$$

where $L_{\text{tri,dis}}$ is the cavity length of the triangular cavity after being disturbed and $R_{M3,dis}$ is the radius of curvature of $C_3$.

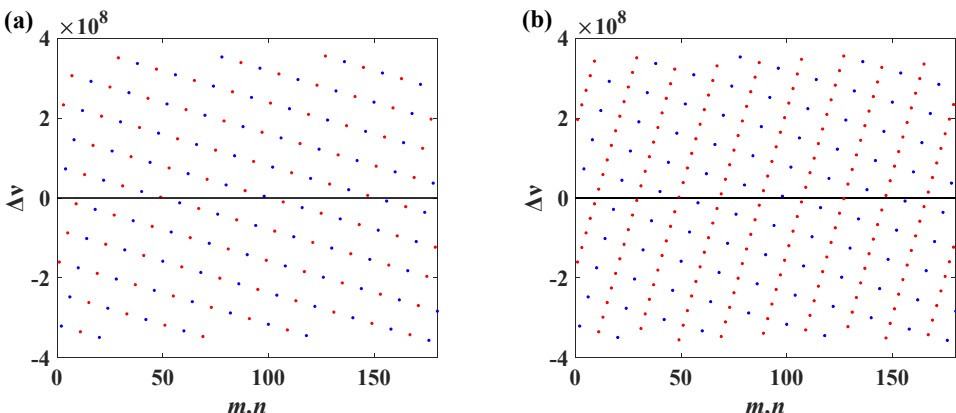

**Figure 3.** Frequency spacing of each order of HGS mode from the fundamental mode in (**a**) a straight cavity and (**b**) triangular cavity. $\Delta v$ represents the frequency spacing of the $n$th- or $m$th-order HGS mode from the fundamental mode in the resonant cavity. Red dots represent odd modes and blue dots represent even modes.

It is assumed that the HGS mode with the smallest frequency spacing from each order of the longitudinal mode is preferentially resonantly coupled. From Equation (6), when $q$ ranges from 1 to 30, the orders of HGS transverse modes closest to the fundamental mode resonance are 4, 9, 13, etc., as shown in Figure 4. The black line represents the ideal $g$-factor for the triangular cavity of 0.58. The blue line indicates the trend in the normalized resonant frequency of the longitudinal mode of order 1–30 when $g$ is varied. The red line represents the trend in the resonant frequency change of the preferentially coupled HGS mode. The intersection of the two curves shows that the HGS transverse mode is coupled to the fundamental mode at this time. The required offset of the $g$-factor when coupling occurs can be seen in Figure 4.

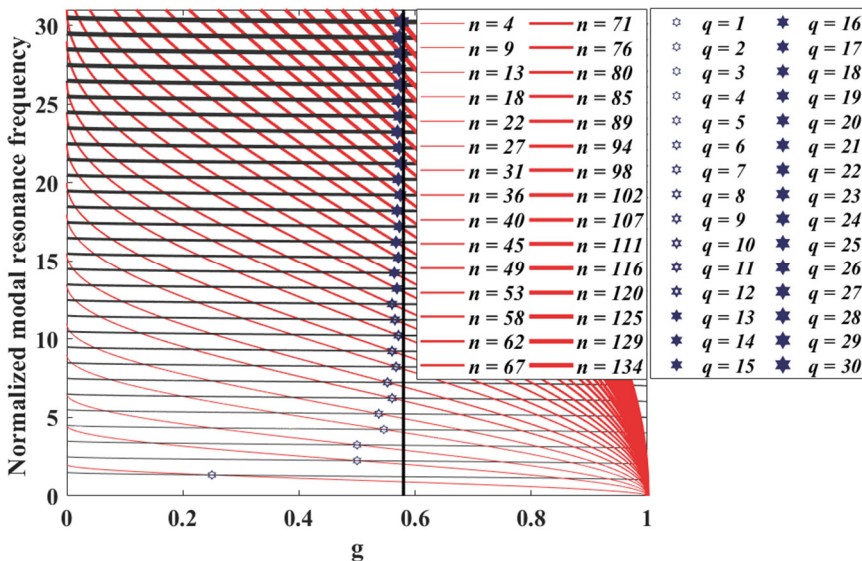

**Figure 4.** Normalized modal resonance frequencies of the $q$th vertical mode (blue line) in the triangular cavity with the corresponding HGS transverse mode (red line) when the $g$-factor is shifted. The vertical line at $g = 0.58$ indicates the undistorted $g$-factor of the triangular cavity. The intersection point indicates that resonant coupling occurred.

As the transverse mode order increases, the degree of $g$-factor shift gradually becomes smaller. Therefore, even if the frequency spacing between the low-order mode and fundamental mode is small, the $g$-factor offset required for resonant coupling to occur is

large, requiring more perturbations from $L$ and $R$. Resonant coupling occurs only when the conditions of $g$-factor shift and frequency shift due to environmental perturbations are satisfied simultaneously.

$$\phi_{Gx} = \arccos\left(\sqrt{1 - \frac{L_{dis}}{R_{dis}}}\right) \tag{16}$$

where $L_{dis}$ is the cavity length of the triangular cavity after being disturbed and $R_{dis}$ is the radius of curvature of $C_3$. In resonant cavities with strong optical energy, such as gravitational wave mode clearers, the environmental perturbation is mainly manifested in changes in $R$ due to thermal deformation. Thermoelastic deformation of the mirror surface is caused by optical energy accumulating in the cavity and being absorbed by the cavity coating. Winkler and Beyersdorf [25,26] provided the change in the radius of curvature $R$ of a mirror induced by thermoelastic deformation at the reflector spot. The perturbation of $R$ is triggered by changes in the sagitta $\delta s_{dis}$ and beam size $\delta \omega_{dis}$ at the cavity mirror.

$$R_{dis} = \frac{(\omega_{cold} - \delta\omega_{dis})^2}{2(s_{cold} - \delta s_{dis})} \tag{17}$$

where $\delta s_{dis}$ and $\delta \omega_{M3,dis}$ at the cavity mirror $C_3$ are given by Equation (18).

$$\begin{aligned} \delta s_{dis} &= \frac{\alpha}{4\pi\kappa}P_{abs} \\ \delta\omega_{M3,dis} &= -\frac{\pi}{2\lambda}\frac{\delta s_{3,dis} \times \omega_{M3}}{[g_1 g_2 (1 - g_1 g_2)]^{1/2}} \end{aligned} \tag{18}$$

where $\alpha$ is the thermal expansion coefficient of the optical substrate, $\kappa$ is the thermal conductivity of the optical substrate, and $P_{abc}$ is the absorbed power of the mirror.

The change in $\Delta\phi_{Gx}$ due to thermal deformation is obtained by differentiating $R$ in Equation (16). The $\Delta R$ leads to a shift of the Gouy phase.

$$\Delta\phi_{Gx} = \frac{1}{2\Delta R_x \sqrt{\Delta R_x - 1}} \tag{19}$$

The thermally induced change in the modal cleaner length $\Delta L$ produces a small modal frequency shift compared to the change in the radius of curvature of the cavity mirror $\Delta R$. Therefore, the thermal deformation is approximated as the change in radius of curvature [15]. CRDS systems are different from gravitational wave mode clearers. The gravitational wave incident light energy is $10^5$ times higher than that of a CRDS system. The thermal deformation will not be the main reason for the shift in the Gouy phase; this is because the $\Delta L$ of the cavity will have a greater effect than $\Delta R$ during a cavity length scan of a CRDS to establish the resonance, or during a cavity pressure scan. Differentiating $L$ for Equation (16) results in:

$$\Delta\phi_{Gx} = \frac{1}{2\sqrt{R_x\Delta L - \Delta L^2}} \tag{20}$$

For a weak energy system such as a CRDS, the $\Delta L$ factor is the main reason for the shift in the resulting Gouy phase. The scattering coupling model cannot distinguish the different mechanisms of action of $\Delta R$ and $\Delta L$. In Section 3.3, we will analyze the mechanism of the effect of $\Delta R$ and $\Delta L$ on modal coupling in detail. All these couplings can be understood as an offset of the $g$-factor.

### 3.3. Coupling Order and Coupling Range of HGS Modes

In existing coupling theory, usually only the frequency shift of the higher-order modes is considered as a condition for resonant coupling [15]. Indeed, it is the HGS modes in the resonant frequency range of the fundamental mode that are likely to be resonantly coupled; however, during our study, we found that the coupling of HGS modes to the fundamental mode was affected by the superposition of several elements, such as diffraction-limited and $g$-factor offset-limited.

When environmental disturbances cause the resonant frequencies of HGS modes to shift at different rates, the frequencies of certain HGS modes coincide with those of the fundamental modes. At this point of resonant coupling, the resonant HGS modes can accumulate high power due to the coherent superposition of the amplitudes of multiple round trips. The HGS-mode frequency-shift range depends on the coupling coefficient $C$ and the loss factor $A$; it has become an industry consensus that the loss of higher order modes is limited by the size of the aperture diaphragm.

When the frequency interval is shifted after environmental perturbation, at this time, if $k$ and $j$ are integers, the HGS mode is resonantly coupled to the fundamental mode.

$$\begin{aligned} \omega_{mq'} - \omega_{0q} &= \frac{m}{\pi} \cos^{-1}\left(\sqrt{g_1 g_{2,dis}}\right) + (q' - q) = k \\ \omega_{nq'} - \omega_{0q} &= \frac{n}{\pi} \cos^{-1}\left(\sqrt{g_1 g_{2,dis}}\right) + \frac{1}{2} + (q' - q) = j \end{aligned} \tag{21}$$

The frequency-shift ranges corresponding to different intensity coupling coefficients $C$ and different higher-order mode losses $A$ are given by Equation (13). The horizontal axis in Figure 4 shows the perturbed phase shift. The vertical axis is the normalized frequency coupling range for (a) different coupling coefficients and (b) different losses.

Through the analysis of Equation (12), we can see that the coupling range is proportional to the coupling strength $C$ and inversely proportional to the loss factor $A$; this means that the coupling offset range of the HGS mode is larger when the loss becomes stronger; this means that in the suppressed higher-order mode scheme with the addition of an aperture, the loss coefficient $A$ of the higher-order mode becomes smaller with the addition of a hard aperture, which will expand the corresponding resonant coupling range. This is shown in Figure 5.

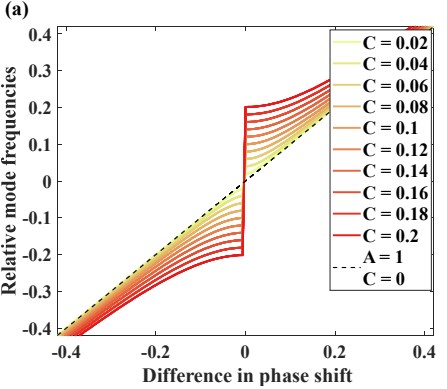 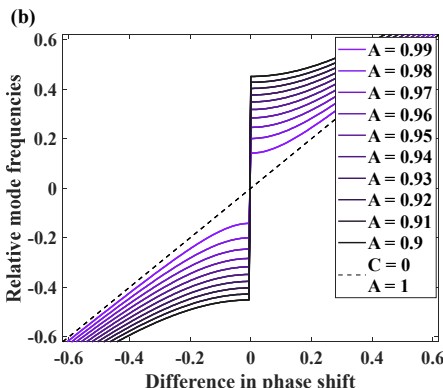

**Figure 5.** Resonant coupling ranges corresponding to different (**a**) intensity coupling coefficients $C$ and (**b**) higher-order mode losses $A$. Solid lines: Relative frequency of the HGS eigenmodes when coupling occurs ($\Delta\nu/FSR$), with the horizontal axis being the perturbed phase shift ($\Delta$ in Equation (12)). Dashed lines: Relative frequency without mode coupling.

The key to resonant coupling is that the frequency shift generated by the HGS mode is within the degeneration range of the fundamental mode; however, this does not mean that all the HGS modes near the resonance point of the fundamental mode are preferentially coupled. When the order of the higher-order modes is too high, there is excessive diffraction loss from the cavity mirror. As the higher-order mode order decreases, the scattering angle of the middle-order mode is larger and the coupling strength is lower than that of the lower-order mode; however, this does not mean that the lower-order mode with the strongest excitation intensity will be coupled preferentially. Figure 6 shows the HGS modes nearest to the resonance points of the different fundamental longitudinal modes. The orders of the HGS modes are the same as those in Figure 3, with orders 4, 9, 13, etc. Taking the 4th-order transverse mode ($n = 4$) as an example, the red line represents the 4th-order transverse mode at the same-order longitudinal mode $\Delta q = 0$, the green line represents the 4th-order transverse mode when $\Delta q = 1$, and the blue line represents the 4th-order transverse mode

when $\Delta q = 2$. The gray line represents the fundamental longitudinal mode coupled with the HGS transverse mode. The black vertical line represents the fundamental $g$-factor ($g = 0.58$) when there is no perturbation. When the transverse mode is a low-order mode ($n < 30$), the $g$-factor requires a large perturbation offset to resonantly couple. Thus, the low-order mode is excited only when the environmental perturbation is large.

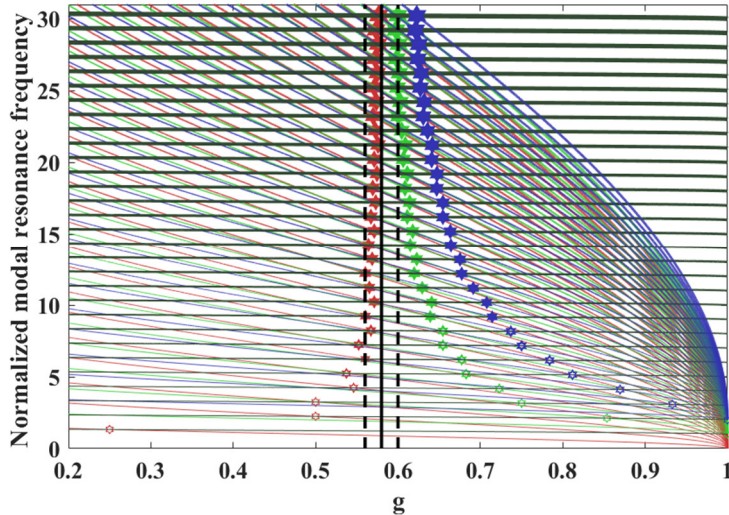

**Figure 6.** Star dots indicate that the fundamental mode longitudinal mode of the triangular cavity (gray line) is resonantly coupled to the HGS transverse mode (colored line) at the nearest resonant frequency. The vertical axes are the respective normalized mode resonant frequencies at the $g$-factor shift. The longitudinal mode $\Delta q = 0$ (red dots), $\Delta q = 1$ (green dots), represents $\Delta q = 2$ (blue dots).

When the environmental perturbation is low, both low-order and high-order modes will be preferentially avoided for resonant coupling. Therefore, we propose the concept of preferential coupling of middle-order modes. A specific case needs to be analyzed with different cavity characteristics. The middle-order mode has lower diffraction loss than the high-order mode, as well as a narrower $g$-factor offset range than the low-order mode. Another condition for preferential coupling of middle-order modes is that there are far more modes within the mode family (the number of modes contained in the same order mode) of the middle-order mode than that of the lower-order mode.

Environmental perturbations will excite HGS modes, and the lower-order mode coupling excited by symmetric misalignment errors will have a greater impact on beam quality. Sayeh conducted much research on the relationship between low-order modes and misalignment mismatch [27]. A triangular cavity attenuates the sensitivity of a straight cavity to symmetry error and helps to distinguish the different coupling mechanisms of the low-order mode from the middle-order mode. The modal filtering and polarization filtering characteristics of a triangular cavity can better determine the coupling order and frequency offset range of the HGS mode. Section 4 presents experiments that were conducted in a CRDS high-finesse triangular cavity with modal resonance coupling.

Authors should discuss the results and how they can be interpreted from the perspective of previous studies and the working hypotheses. The findings and their implications should be discussed in the broadest context possible. Future research directions may also be highlighted.

## 4. Experiments

The experiments were divided into two stages. The first stage verified the injection and coupling of low-order modes, and the second stage explored the coupling and cancellation of middle-order modes. The injection of low-order modes is divided into two types for multimode fiber and misalignment excitation; it is obvious that a moderate amount of

misalignment introduces less spectral noise and will be more favorable to the detection of ring-down phenomena [16]. Higher-order modes are introduced by environmental perturbations, and resonant coupling with the fundamental mode ensues. Coupling of higher-order modes was found when conducting the PZT (piezoelectric ceramic transducer) cavity length scan and cavity gas pressure scan. Both the low-order and high-order modes excited in a CRDS cannot be determined using a conventional mode analyzer because the transmission energy of a CRDS high-finesse cavity can only be detected by a near-infrared high-gain photomultiplier device. We exploited the time-domain properties of the high-finesse ring-down cavity to determine the excitation of higher-order modes.

Using the time domain method to detect multimode coupling was first adopted in the field of CRDS [23]. When the fundamental mode is single-mode coupled, the ring-down curve is fitted to a single-e exponential curve (Equation (22a)). When the cavity lifetime decreases during the cavity length scan, the ring-down curve at this time is found to fit a double-e exponential model (Equation (22b)). The double-e exponential model at this point is considered to represent coupling of the higher-order mode to the fundamental mode.

$$I_1(t) = A + B\exp(-kt) \tag{22a}$$

$$I_2(t) = A + B_1\exp(-k_1 t) + B_2\exp(-k_2 t) + 2\sqrt{B_1 B_2} \\ \times \exp(\tfrac{-k_1 t}{2})\exp(\tfrac{-k_2 t}{2})\cos(2\pi\Delta\nu t + \Delta\phi) \tag{22b}$$

where $I(t)$ is the detector voltage, $A$ is the detector DC offset, $B$ is the amplitude at the beginning of the ring-down, $k$ is the cavity ring-down rate, $\Delta\nu$ is the frequency difference between the two modes, and $\Delta\phi$ is the initial phase difference. The model uses chi-squared ($\chi^2$) tests to determine the deviation between observed values and the ring-down fitted curve. When the $\chi^2$-value is significant, the intracavity ring-down can be judged as single-mode or multimode coupled ring-down; however, the premise of this test is that the ring-down curve requires a large amount of data fitting. The ring-down time constant of the system is in the order of 10 μs. To obtain more than 1000 data signals in a single decay period, the sampling frequency should be > 100 MHz.

$$\chi^2 = \sum_{i=0}^{n} \frac{(A(i) - I(i))^2}{I(i)} \tag{23}$$

Figure 7 shows the instrument used to demonstrate the mode coupling experiment. A narrow linewidth (2 MHz) single-mode laser emitted from a light source (DFB, EP1653-DM-B) was divided into a reference optical path and a main optical path by a beam splitter. The role of the reference optical path is described in the second part of this section. After the main optical path passed through the polarizing beam splitter, the single-polarized light entered the ring-down cavity through a matching mirror and alignment mirrors $M_1$ and $M_2$ [28]. A column lens was added to the matching mirror to eliminate the effect caused by the image dispersion of the triangular cavity. The initial alignment of the system and the introduction of low-order modes were achieved via alignment mirrors $M_1$ and $M_2$. The ring-down cavity was made of invar and the cavity had inlet and exhaust ports, with the gas flow rate determined by a mass flow controller. A gas filter was installed in the inlet port to remove particles that could become a source of ring-down rate noise. The cavity was equipped with both a platinum resistance thermometer (BD-WZP-PT100) and a pressure transmitter (HPT700) to measure the temperature and pressure of the gas inside the cavity. The round-trip length of the attenuation cavity was $L = 420$ mm, the angle between $C_1$ and $C_2$ was $88.542°$, and $C_3$ had a radius of curvature of 1 m. The cavities $C_1$, $C_2$, and $C_3$ met the impedance-matching conditions of 99.97%, 99.9%, and 99.997% reflectance at 1653 nm, respectively. The other single polarizer of the polarizing beam splitter used the classical PDH method and incorporated a frequency sideband to mix with the outgoing cavity light, and the PZT (PZT, P-841.10) scanning cavity length was controlled by a locking servo mechanism to lock the center frequency of the light source at the cavity fundamental mode resonance frequency. An AOM (Acousto-optical Modulators) was used to turn off

the laser source after the light intensity reached the threshold value; it is important to note that the piezoelectric element attached to the end mirror needed to be calibrated before the experiment to eliminate hysteresis. Finally, the light emitted from the ring-down cavity was focused on the detector through a short-focal-length lens. The light signal was detected using an InGaAs Switchable Gain Detector (Thorlab PDA10CS2) and its output was fed to a digital oscilloscope (MSO, MSO6104A).

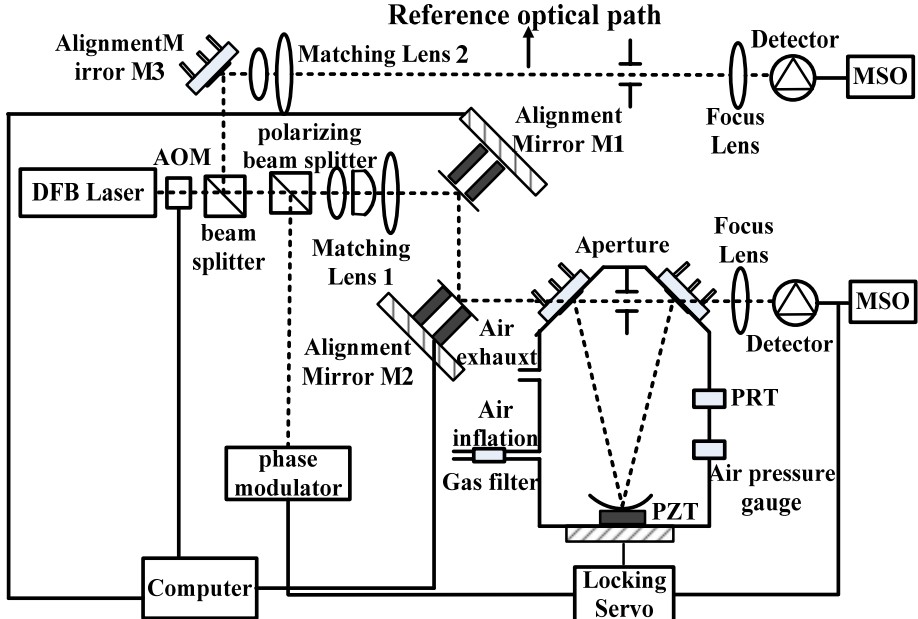

**Figure 7.** Structural diagram of the ring-down cavity HGS-mode resonance coupling experimental setup.

### 4.1. Low-Order Mode Coupling Experiment

After optical path alignment was completed and the fundamental mode resonance was established, the introduction of the low-order mode was completed by alignment mirrors $M_1$ and $M_2$; this was used to measure the coupling between the low-order mode and fundamental mode. The angular tilt of the alignment mirrors was controlled by PZT. The pure angular tilt between the incident beam and cavity beam was controlled by $M_1$, which was close to the input mirror. Optical axis translation was achieved by the combined motion of $M_1$ and $M_2$. To prevent some of the light from exceeding the detection range at excessive tilt angles, the tilt angle ($\alpha_v$) range was limited to $\pm 0.3°$ and the axis translation ($\varepsilon_v$) range was $\pm 0.3$ mm [29]. Where $v$ is the direction $x/y$. The low-order modes were introduced mainly by tilt injection.

The mode coupling experiment was a cavity experiment conducted without the addition of the gas to be measured. The cavity was filled with pure nitrogen and the cavity length was scanned by applying a voltage to the cavity PZT, which scanned over a range of 5 FSR. During scanning, the laser was current-modulated to control the scanning wavelength at each measurement point to approximately 1.1 FSR of the cavity, ensuring that the laser resonated with the cavity's fundamental mode at each cavity length. The ring-down curves at each measurement point were fitted by the least-squares method. The last 5 μs signal was measured before fitting to determine the noise of the detector, since the signal value at this point was already below the noise value, and this noise was subtracted from the previous points [23].

During the cavity-length scan of the triangular cavity, the cavity ring-down experiments revealed non-periodic cavity lifetime anomalies. The ring-down times of these anomalies were lower than the normal cavity lifetimes. The goodness of fit was measured

in terms of $\chi^2$-values, with the $\chi^2$ at the anomalies being significantly larger than that with the normal cavity lifetime. A double-e exponential curve fitted these anomalies well.

To demonstrate whether the multimode coupling at this point was introduced by the excitation of the lower order modes, the cavity was scanned to the normal and abnormal points of the cavity lifetime in the experiment. The resonance was established again after adding the tilt factor (first-order mode excitation enhancement) via tilt mirror $M_1$ in these two cases. Figure 8a shows the ring-down curve of the PZT scanning cavity length vs the normal cavity lifetime after adding misalignment error, and Figure 8b shows the curve fitted to the cavity lifetime anomalies after adding misalignment error. Figure 8 shows that the addition of misalignment error does not affect the shape of the fitted ring-down curve. Figure 8a shows that the ring-down curve can still be fitted as a single-e exponential curve after adding misalignment, while Figure 8b shows that when the cavity length is scanned to the ring-down curve anomaly, the double-e exponential model remains after adding the misalignment error and only the ring-down time is reduced. The above experiments demonstrate that multimode coupling in the ring-down cavity is not due to the injection of low-order modes.

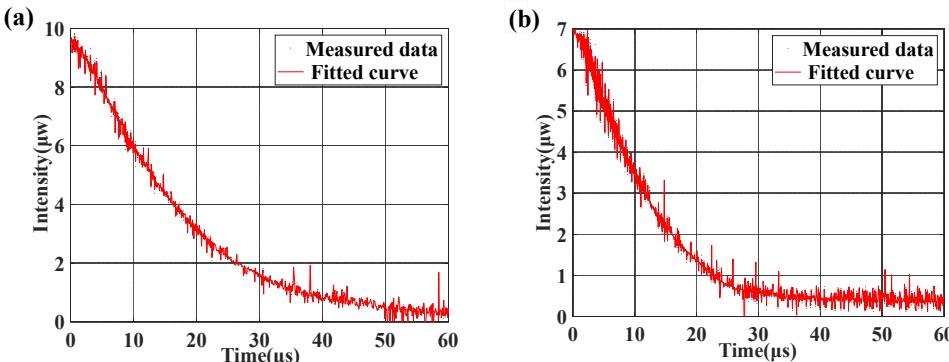

**Figure 8.** Fundamental mode ring-down curves fitted after adding misalignment with cavity lengths scanned to the (**a**) normal and (**b**) abnormal cavity lifetimes, with misalignment (by tilt mirror).

*4.2. Middle-Order Mode Coupling Experiment*

In order to determine the order of the HGS modes resonantly coupled to the fundamental mode, a progressively smaller-diameter aperture diaphragm was added to the cavity. Then, cavity ring-down experiments were performed under different cavity aperture conditions. These experiments confirmed the correlation between cavity lifetime reduction and multimode coupling, which is consistent with the phenomenon observed by Huang in his experiments [23]. Figure 9 shows the distribution of the measured cavity lifetimes for 1000 ring-down events. Points that did not conform to the fitted exponential curve (caused by random errors) were removed. The probability distributions of cavity lifetimes for apertures of 10 mm, 8 mm, and 6 mm were compared. The cavity lifetime anomalies that did not fit the single-e exponential ring-down curve were largely eliminated as the aperture diameter decreased. The following section analyzes why the non-e exponential ring-down disappeared when the aperture diameter was reduced to 6 mm. At the same time, the reduction in aperture diameter led to a weakening of the system's optical energy, which explains the reduction in cavity lifetime at the 6 mm aperture diameter.

The addition of a reference optical path eliminates detector performance degradation with the addition of an aperture. The comparison optical path designed in the experiment had the same optical path characteristics and beam waist as the main optical path. The reference optical path and main optical path were added with the same aperture diaphragm at the same time during the experiment. The weakening of light energy due to the addition of $\geq$6 mm diaphragms to the reference optical path was the same as that in the main optical path.

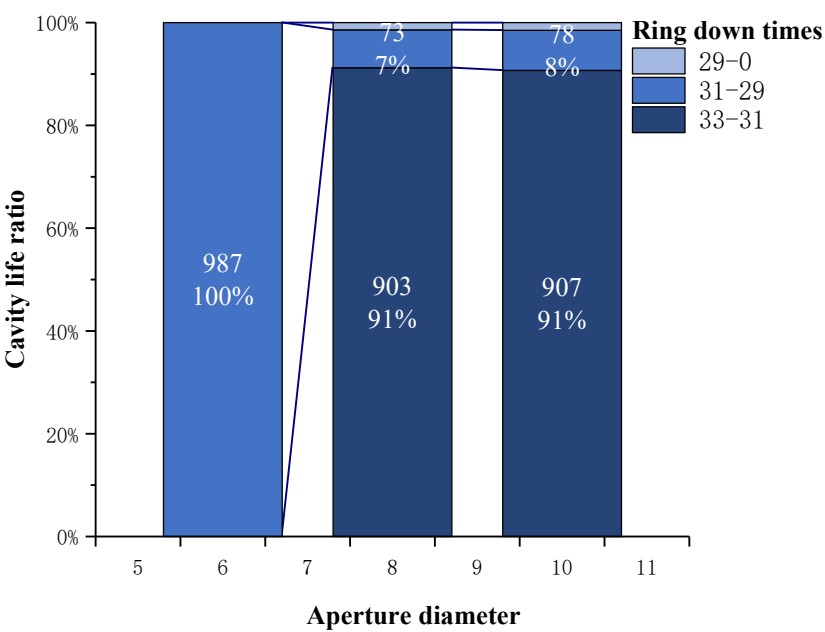

**Figure 9.** Distribution of cavity lifetimes during cavity length scanning with aperture diaphragms of 6 mm, 8 mm, and 10 mm diameter added to the cavity.

## 5. Further Discussion

Good agreement was obtained between repeated cavity length scans in both phases of the experiment. We have reason to believe that in a narrow linewidth system such as a CRDS, excitation of the lower-order mode does not directly resonantly couple with the fundamental mode. Taking the first-order mode as an example, the free spectral range of the resonant cavity was 714.3 MHz. Then, the horizontal and vertical transverse mode spacings in the two planes of the triangular cavity can be calculated by Equation (6) as 160.3 MHz and 517.45 MHz, respectively. Such a transverse mode spacing is much larger than the range of the system's frequency shift.

The frequency drift of the transverse mode of the HGS mode that can be coupled to the fundamental mode should be within the resonance range of the fundamental mode. Equation (14) can be used to calculate the frequency spacing between each order of HGS mode and the fundamental mode; it can be calculated that, in the *x-z* plane when $m + n = 49$, 98 and 107, the frequency spacings are 2.6, 5.2 and −8.9 MHz, respectively, which are close to the linewidth of the light source. In the *y-z* plane, the frequency spacings are 5.8, 5.2, and 0.55 MHz when $m + n = 29$, 98, and 127, respectively. The inference in Section 2 shows that the transverse mode spacings of the same order in the two planes of the triangular cavity are different, but the transverse mode spacings of the two planes are both 5.2 MHz when $m + n = 98$. The beam waist radius of the 98th-order mode was calculated to be 6.7 mm by $\sqrt{m + n + 1}\omega_b$; this reasonably explains the disappearance of the non-e exponential ring-down curve when a 6 mm cavity aperture was added; it can be assumed that the aperture increases the diffraction loss of the 98th-order mode at this time, thus reducing its cross-coupling with the fundamental mode. The 29th and 49th orders do not have persistent cross-coupling with the fundamental mode because they have different resonant frequencies in different planes, with a difference of $FSR/2$. The 98th-order mode of the triangular cavity is an even order mode, and the modes can also be coupled to each other and then resonate with the fundamental mode. By the same principle, TEM01 and TEM10 can be coupled with each other in a straight cavity [16]. In a triangular cavity, only even-order modes have this property.

The coupling range of the 98th-order mode is 5.2 MHz, which is much higher than the 10 kHz linewidth in the cavity. Since high-finesse cavities are more susceptible to frequency degenerate mode coupling than low finesse cavities, the frequency shift of the

HGS mode is proportional to the finesse, resulting in a larger frequency shift for higher finesse cavities [15]; furthermore, the inclusion of the aperture enhances the loss of the HGS mode. The analysis in Section 3 shows that as the loss factor A becomes smaller (i.e., the loss becomes larger), the coupling range increases.

## 6. Conclusions

In this paper, the mechanism of resonant coupling between HGS modes and fundamental modes in resonant cavities was investigated; it was found that different environmental perturbations lead to coupling between modes triggered by both the $\Delta R$ and $\Delta L$ factors. The conditions for the occurrence of resonant coupling are not only subject to the single factor of frequency drift range but also need to consider the diffraction limit, $g$-factor drift range. The concept of middle-order modes has been proposed for the first time, and the middle-order modes will be preferentially coupled. In the experiment, the coupling and excitation of HGS modes were verified using a time-domain approach based on the unique modal filtering and polarization filtering characteristics of the triangular cavity. The middle-order modes are the main source of cavity lifetime anomalies in the CRDS system. Finally, the order and frequency coupling range of the middle-order modes were analyzed. The time-domain analysis method provides new ideas for conducting experiments when a modal analyzer cannot be used in low-energy systems.

**Author Contributions:** Conceptualization, Y.R.; moreover C.H.; methodology, Y.R.; software, Y.R.; moreover C.Y.; validation, C.Y.; moreover C.H.; formal analysis, Y.R.; investigation, Y.R.; moreover X.Z.; resources, C.H.; writing—original draft preparation, Y.R.; writing—review and editing, Y.R.; moreover X.Z.; funding acquisition, C.H.; moreover C.Y. All authors have read and agreed to the published version of the manuscript.

**Funding:** This work was supported by the National Natural Science Foundation of China, grant number 61805235, grant number 61905241 and grant number 61875192.

**Conflicts of Interest:** The authors declare no conflict of interest.

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
