# Peer review of "Resonant Coupling of Hermite-Gaussian Transverse Modes in the Triangular Cavity of a Cavity Ring-down Spectroscope"

_photonics, doi:10.3390/photonics9090595_

Round 1
Author Response
Response to Reviewer 1 Comments
Point 1: The authors present in this paper a unified explanation of the different types of resonant coupling and also provide a new approach to resonant coupling experiments performed in high finesse resonant cavities. The analysis method used by the authors in the time domain presents new ideas for conducting experiments when a modal analyzer cannot be used in low-energy systems.
The abstract is very explicit both in terms of methods and results. The objects were presented in a well-structured way. The analysis method is appropriately described. The body of the paper is clear and It’s correctly written. The conclusions are supported by the results.
The work presented remains original since, in terms of plagiarism assessment, only about 23% of this document consists of text more or less similar to the content of the sources considered most relevant by iThenticate. I believe this is an interesting contribution and I recommend publication in Photonics in its present form. For language, I don't feel qualified to judge the English language
and style.
Response 1:
Dear reviewer,
Thank you for your comments concerning our manuscript. We are deeply grateful for our paper reviewed by experts in this field. All of the comments are really valuable and helpful to improve our paper; as well as important to guide our future researches.

Reviewer 2 Report
The article by Ren and co-workers provides a concise description of their analysis of resonant couplings of transverse modes of laser pulses in a triangular ring-down cavity. Various mechanisms are examined as causes for degradation of the laser-mode quality from the ideal TEM00 mode. Together with applications to the experimental observations, the theory is well inspected and concretely understood. Although the theoretical part needs a book-chapter volume for comprehensive description, it is worth published in Photonics for a record of the author’s attainment.
Before finalizing the manuscript, the following issues are better to be rephrased for readers who are not very familiar to the subject.
Lines 113 and 115: The explanation “w(q(z(j))) is the beam radious of the fundamental mode” appears twice and can be deleted one of them. Rather, an explanation of w0 is needed here.
For Figure 1, it is helpful to explain explicitly the relation between the directions of transverse modes (x, y), (m, n) and the directions, parallel (in-plane) or perpendicular (out-of-plane), to the plane of the triangular cavity.
Line 147: The concave mirror is characterized by its curvature, R. For experimentalists, it is familiar to be given here a relation between the focal length of the mirror, f, and the parameter, R.
Line 227: For the Equation (13), is energy conservation fulfilled? In other words, is it normalized by the energy of the incident light? Or, does it provide merely the ratio of the coupled modes to the fundamental mode, as the perturbation theory provides?
Line 295: The meaning of the subscript “dis” is better to be explained.
Line 302: The parameter, delta_s_dis, appears twice. The latter maybe replaced with delta_w_dis.
Line 420: The function, y(t), does not appear in the equations (22a) and (22b). It may be replaced with I(t).
Line 422: The parameter, delta_f, may be replaced by delta_phi.
In Figure 7, the position of the aperture is better shown/indicated.
Line 460: Is it intentionally written “Bulleted lists look like this:” ?
Line 468: The meaning of the subscript “v” is better to be explained.
Author Response
Response to Reviewer 2 Comments
Point 1: Lines 113 and 115: The explanation “w(q(z(j))) is the beam radious of the fundamental mode” appears twice and can be deleted one of them. Rather, an explanation of w0 is needed here.
Response 1: Thank you very much for your suggestion, it is very meaningful to improve the quality of the paper. We have made corresponding changes in the revised manuscript. Lines 113 and 115 have been changed to: ω0 is the beam radius of the fundamental mode, ω(q(z(j))) is the beam radius of the fundamental mode at position z(j),
Point 2: For Figure 1, it is helpful to explain explicitly the relation between the directions of transverse modes (x, y), (m, n) and the directions, parallel (in-plane) or perpendicular (out-of-plane), to the plane of the triangular cavity.
Response 2: Thank you very much for your suggestion, it is very meaningful to improve the quality of the paper. We have made corresponding changes in the revised manuscript. The following has been added above Figure 1:
x, m represent the direction parallel to the plane of the triangular cavity. y, n represent the direction perpendicular to the plane of the triangular cavity.
Point 3: Line 147: The concave mirror is characterized by its curvature, R. For experimentalists, it is familiar to be given here a relation between the focal length of the mirror, f, and the parameter, R
Response 3: Thank you very much for your suggestion, it is very meaningful to improve the quality of the paper. We have made corresponding changes in the revised manuscript. We give the relationship between the mirror focal length f and the parameter R on line 147, f=R/2
Point 4: Line 227: For the Equation (13), is energy conservation fulfilled? In other words, is it normalized by the energy of the incident light? Or, does it provide merely the ratio of the coupled modes to the fundamental mode, as the perturbation theory provides?
Response 4: Thank you very much for your suggestion, it is very meaningful to improve the quality of the paper. We have made corresponding changes in the revised manuscript. Equation (13) only provides the ratio of the coupled mode to the fundamental mode
Point 5: Line 295: The meaning of the subscript “dis” is better to be explained.
Response 5: Thank you very much for your suggestion, it is very meaningful to improve the quality of the paper. We have made corresponding changes in the revised manuscript. We added on line 295: Where Ldis is the cavity length of the triangular cavity after being disturbed and Rdis is the radius of curvature of C3.
Point 6: Line 302: The parameter, delta_s_dis, appears twice. The latter maybe replaced with delta_w_dis.
Response 6: Thank you very much for your suggestion, it is very meaningful to improve the quality of the paper. We have made corresponding changes in the revised manuscript. We changed line 302 to: Where and at the cavity mirror C3 are given by Eq. (18).
Point 7: Line 420: The function, y(t), does not appear in the equations (22a) and (22b). It may be replaced with I(t).
Response 7: Thank you very much for your suggestion, it is very meaningful to improve the quality of the paper. We have made corresponding changes in the revised manuscript. We changed line 420 to: Where I(t) is the detector voltage,
Point 8: Line 422: The parameter, delta_f, may be replaced by delta_phi
Response 8: Thank you very much for your suggestion, it is very meaningful to improve the quality of the paper. We have made corresponding changes in the revised manuscript. We changed line 422 to: ΔÏ•is the initial phase difference.
Point 9: In Figure 7, the position of the aperture is better shown/indicated.
Response 9: Thank you very much for your suggestion, it is very meaningful to improve the quality of the paper. We have made corresponding changes in the revised manuscript. We changed Figure 7 to:
Point 10: Line 460: Is it intentionally written “Bulleted lists look like this:” ?
Response 10: Thank you very much for your suggestion, it is very meaningful to improve the quality of the paper. We have made corresponding changes in the revised manuscript. We deleted the original content " Bulleted lists look like this:" on line 460.
Point 11: Line 468: The meaning of the subscript “v” is better to be explained.
Response 11: Thank you very much for your suggestion, it is very meaningful to improve the quality of the paper. We have made corresponding changes in the revised manuscript. We added "Where ν is the direction x/y" on line 468
